# Characterization of *Brucella abortus* Mutant A19mut2, a Potential DIVA Vaccine Candidate with a Modification on Lipopolysaccharide

**DOI:** 10.3390/vaccines11071273

**Published:** 2023-07-21

**Authors:** Hosny Ahmed Abdelgawad, Zhengmin Lian, Yi Yin, Tian Fang, Mingxing Tian, Shengqing Yu

**Affiliations:** 1Shanghai Veterinary Research Institute, Chinese Academy of Agricultural Sciences (CAAS), Shanghai 200241, China; hosnyahmed@vet.aswu.edu.eg (H.A.A.); lian_zm@126.com (Z.L.); yinyisonia@126.com (Y.Y.); ft18052078884@163.com (T.F.); 2Department of Microbiology and Immunology, Faculty of Veterinary Medicine, Aswan University, Aswan 81528, Egypt; 3Jiangsu Key Laboratory for High-Tech Research and Development of Veterinary Biopharmaceuticals, Jiangsu Agri-Animal Husbandry Vocational College, Veterinary Bio-Pharmaceutical, Taizhou 225309, China

**Keywords:** *Brucella abortus*, vaccine, differential diagnosis, lipopolysaccharide, modification

## Abstract

Background: *Brucella abortus* is the main causative agent for bovine brucellosis. *B. abortus* A19 is a widely used vaccine strain to protect cows from *Brucella* infection in China. However, A19 has a similar lipopolysaccharide (LPS) antigen to that of the field virulent *Brucella* strain, whose immunization interferes with the serodiagnosis of vaccinated and infected animals. [Aim] To develop a novel *Brucella* DIVA vaccine candidate. Study design and methods: The *B. abortus* mutant A19mut2 with the formyltransferase gene *wbkC* is replaced by an acetyltransferase gene *wbdR* from *E. coli* O157 using the bacterial homologous recombination technique, generating a modified O-polysaccharide that cannot induce antibodies in mice against wild-type *Brucella* LPS. The biological phenotypes of the A19mut2 were assessed using a growth curve analysis, agglutination tests, Western blotting, and stress resistance assays. Histopathological changes and bacterial colonization in the spleens of vaccinated mice were investigated to assess the residual virulence and protection of the A19mut2. Humoral and cellular immunity was evaluated by measuring the levels of IgG, IgG subtypes, and the release of cytokines IFN-γ and IL10 in the splenocytes of the vaccinated mice. ELISA coated with wild-type LPS can distinguish mouse antibodies induced by A19 and A19mut2 immunization. Results: The A19mut2 showed a decreased residual virulence in mice, compared to the A19 strain, but induced significant humoral and cellular immune responses, as the A19 immunization did. The protection efficacy of A19mut2 immunization against *B. abortus* S2308 Nal^R^ infection was similar to that of A19 immunization. Conclusion: The A19mut2 has potential as a novel DIVA vaccine candidate in the future.

## 1. Introduction

Brucellosis is an important recurrent zoonosis that causes significant economic losses, including in terms of livestock, and poses a substantial public health risk. Brucellosis is caused by a pathogen known as *Brucella*, which lacks spores and is a Gram-negative coccobacillus. To decrease these high economic losses, the production of an effective veterinary vaccine would be an effective strategy to control *Brucella* in humans and animals [1,2].

Vaccination is now the most cost-effective method of preventing and controlling brucellosis. In China, the *B. abortus* vaccine A19 has been used extensively to protect cattle, but the antibody response, especially induced by the O-polysaccharide (O-PS) of LPS, hinders serological diagnoses, including serum agglutination tests, Rose Bengal plate tests, complement fixation tests, and enzyme-linked immunosorbent assays (ELISAs); thus, it is difficult to distinguish between infected and vaccinated animals [3,4]. There is no doubt that the limitation of the A19 vaccine significantly affects ongoing efforts to eliminate and control brucellosis. However, the use and development of gene-deleted vaccines could be an effective strategy to overcome the interference with serological diagnoses [3,5].

Recently, a *B. melitensis* Rev1 vaccine strain with the deletion of the formyltransferase gene *wbkC* associated with O-PS synthesis and tagged with acetyltransferase *wbdR* from *E. coli* elicited antibodies that could be distinguished from those evoked by wild-type strains, was relatively attenuated, and provided protection against *B. ovis* [6]. This ingenious strategy could effectively solve the shortcomings of residual virulence and the differential diagnosis of vaccines. However, related strategies have not been used and evaluated for *B. abortus* vaccines, which motivated us to conduct relevant studies in *B. abortus* vaccine strain A19.

In this study, the *wbkC* gene in *B. abortus* A19 was replaced by the *wbdR* gene from *E. coli* O157, generating an O-PS-modified mutant strain, A19mut2. Our findings suggest that the A19mut2 exhibits potential as a DIVA (differentiating infected from vaccinated individuals) vaccine candidate against brucellosis.

## 2. Materials and Methods

### 2.1. Ethics Statement

The experimental protocol was approved by the Shanghai Veterinary Research Institute, the Chinese Academy of Agricultural Sciences (CAAS), Shanghai, China (permit no. SHVRI-SD-2019-105). The protocol was strictly following the Care and Use of Laboratory Animals of the Institutional Animal Care and Use Committee of Shanghai Veterinary Research Institute, CAAS. 

### 2.2. Brucella Strains and Cultural Conditions

*B. abortus* vaccine strain A19 and wild-type strain S2308 were obtained from the Chinese Veterinary Culture Collection Center (CVCC, Beijing, China). *B. abortus* nalidixic acid-resistant strain 2308 Nal^R^ was derived from the S2308 by induction of nalidixic acid [7]. The S2308 strain was used to prepare the heat-killed antigens. The S2308 Nal^R^ strain was used as a challenge to determine the protection of vaccination. *B. abortus* rough-type strain RB14 (*B. abortus* strain 2308 with deletion of *rfbE* gene) was constructed in our previous study [8]. *Brucella* strains were cultured in tryptic soy broth (TSB, Difco, NJ, USA) or tryptic soy agar (TSA) at 37 °C with 5% CO_2_. Experiments dealing with live *B. abortus* strains were all performed in a biosafety level 3 laboratory facility at the Chinese Academy of Agricultural Sciences. *E. coli* strain O157 (passaged and stocked in our lab) and DH5α (TIANGEN Biotech Co., Ltd., Beijing, China) were grown in Luria-Bertani medium. When appropriate, 50 μg/mL of kanamycin or 15 μg/mL of nalidixic acid (Sigma-Aldrich Corporation, St. Louis, MO, USA) were added. All strains and plasmids used in this study are listed in Table 1.

### 2.3. Mutant Construction

A suicide plasmid was constructed as previously described with modifications [9]. Briefly, a 2795 bp fragment, including *wbkC* gene and its upstream and downstream fragments, was amplified by PCR from the genome of *B. abortus* A19 strain using primers WbkC-F and WbkC-R. The fragment was cloned into PCR-linearized pKB plasmid using a seamless cloning and assembly kit (ClonExpress II One Step Cloning Kit, Vazyme, Nanjing, China), and then the recombinant plasmid was further linearized by inverse PCR using primers rWbkC-F and rWbkC-R to exclude the *wbkC* gene. Moreover, *wbdR* gene containing its promoter region was amplified from *E. coli* O157 by PCR using primers WbdR-F and WbdR-R. The recovered *wbdR* fragment was cloned into inverse PCR-linearized recombinant plasmid to replace *wbkC* gene using the seamless cloning and assembly kit as described above. The suicide plasmid pKB-ΔwbkC::wbdR was propagated in *E. coli* DH5α cells and used to construct the mutant. A19mut2 mutant was constructed by allelic replacement using a two-step strategy as previously described [9].

### 2.4. Bacterial Growth Determination

Bacterial growth of A19 and A19mut2 strains in the TSB was measured with determination of the optical density at 600 nm (OD_600_) using an Eppendorf BioSpectrometer basic (Eppendorf, Hamburg, Germany) as described [9]. Briefly, bacterial cells were washed twice, the OD_600_ value was adjusted to 0.1 (5 × 10^8^ CFU/mL) in TSB, and then they were cultured at 37 °C at 200 rpm to generate growth curves. The OD_600_ absorbance of aliquots was measured every 4–8 h, and the records were used to draw the growth curve.

### 2.5. Acridine Yellow Agglutination and Heat Agglutination Tests

Bacterial cultures at early stationary stage were washed twice with PBS and concentrated 10-fold by centrifugation. Then 10 μL bacterial suspension was evenly mixed with 10 μL 1% acridine yellow solution and incubated for five mins to determine the acridine yellow agglutination. The early stationary stage of the *Brucella* culture suspension was incubated for 2 h in a water bath at 80 °C to evaluate the heat agglutination. The rough-type *Brucella* strain RB14 was used as a positive control.

### 2.6. LPS Extraction

A19 and A19mut2 strains were cultured to exponential phase in TSB. Bacterial cells were collected by centrifugation, and crude LPS was extracted using proteinase K-digested whole-cell lysates [10,11]. Samples were loaded on 12.5 % polyacrylamide gels for SDS-PAGE, and a silver staining assay was performed as previously described [9]. To prepare LPS for ELISA, we isolated LPS using hot phenol extraction as described previously.

### 2.7. SDS-PAGE and Western Blotting

The extracted LPS was mixed with 5 × SDS loading buffer (Beyotime, Suzou, China) and boiled for 10 min. Samples were loaded for SDS-PAGE using 12.5 % resolving gels. Western blotting was performed as previously described [9]. Rabbit anti-*Brucella* antibody or mouse anti-O-PS monoclonal antibody 51C was used as the primary antibody, and IRDye 680RD Goat anti-Mouse IgG (H+L) or IRDye 800CW Goat anti-Rabbit IgG (H+L) (LI-COR Biosciences, Lincoln, NE, USA, 1: 20,000 diluted) was used as the secondary antibody. Images were acquired by scanning using Odyssey Imaging System (LI-COR).

### 2.8. Stress Resistance Assay

Stress resistance assays were performed as described previously [9]. Resistance to H_2_O_2_, polymyxin B, sodium nitroprusside (SNP), and acidic peptone water (1 g/L Tryptone; 5 g/L NaCl; pH = 7.0, 6.5, 5.5, or 4.5) was performed to determine the sensitivity of the *Brucella* strains to oxidative stress, cationic bactericidal peptides, nitrosative stress, and the acid tolerance, respectively. 

### 2.9. Virulence Test

Virulence test was performed as described previously [12]. The OD_600_ value of bacterial cultures was adjusted to 1.0 (5 × 10^9^ CFU/mL). Bacterial suspension was serially diluted 10-fold to 5 × 10^6^ CFU/mL. Groups of BALB/c mice (*n* = 20 per group) were inoculated intraperitoneally with 1 × 10^6^ CFU of A19 or A19mut2 in 0.2 mL of PBS, and PBS was injected as a negative control. At 2-, 4-, 6- and 8-weeks post inoculation (WPI), five mice from each group were sacrificed, and their spleens were aseptically removed and homogenized in PBS. Ten-fold serial dilutions of spleen homogenates were plated on TSA plates and incubated for 3–5 days at 37 °C. The CFUs were enumerated to evaluate colonization efficiency of each strain in mice. The results are presented as the mean ± standard deviation (SD) of the log10 (log) CFU/spleen for each group.

### 2.10. Histopathological Examination

Histopathological examination of spleens and livers from mice inoculated with A19, A19mut2, or PBS was completed by Wuhan Servicebio Co., Ltd. (Wuhan, China) as previously described [13]. Briefly, the samples were fixed by 4% paraformaldehyde in properly fixed condition, and then trimmed, dehydrated, embedded, sectioned, stained by hematoxylin and eosin, sealed, and finally microscopically qualified. Tissue sections were browsed under the microscope and observed in detail at different magnifications. Basic pathological changes in the sections were described in writing, such as congestion, bruising, hemorrhaging, edema, degeneration, necrosis, hyperplasia, fibrosis, mechanization, granulation tissue, inflammatory changes, etc. Typical lesion sites were imaged and are identified with arrows in images.

### 2.11. Evaluation of Antibody Response in Mice Serum

This experiment was performed as previously reported [14]. Mice (*n* = 5 per group) were bled from the orbital sinus after anesthesia of isoflurane, and the sera were kept at minus 70 °C until testing. ELISA was used to test the levels of IgG and IgG subclasses of antibodies. Briefly, ELISA plates were coated with 25 μg/well heat-killed and sonicated *Brucella* in carbonate buffer (pH = 9.6) overnight at 4 °C. Triplicate serum samples with 200-fold dilution were added as the primary antibody. Goat anti-mouse IgG (Thermo Fisher, Waltham, MA, USA), IgG1, or IgG2a (Abcam, Cambridge, MA, USA) conjugated to horseradish peroxidase was used as the secondary antibody. Moreoever, ELISA was used to detect the antibodies against LPS, and the microplate was coated by the hot-phenol-extracted LPS with 0.1 μg per well in carbonate buffer (pH = 9.6) overnight at 4 °C. Goat anti-mouse IgG was used as the secondary antibody. All the procedures were carried out according to the previous report [14,15].

### 2.12. Detection of Cytokine in Supernatants of Splenocytes

Splenocytes were prepared from the vaccinated and PBS mice as described previously [12]. At 6 WPI, the splenocytes were isolated, suspended in RPMI1640 (Thermo Fisher) with 10% FBS (BI), and seeded at a density of 2 × 10^5^ cells/well in 96-well cell culture plate, and then stimulated with 1 × 10^8^ CFU heat-killed S2308 strain or RPMI1640. At 3 days post stimulation, the supernatants were collected, and the levels of the IFN-γ and IL-10 were detected using ELISA kits according to the manufacturer’s instructions (Bio-Legend, San Diego, CA, USA).

### 2.13. Protective Efficacy Analysis

Six-week-old female BALB/c mice were randomly divided into three treatment groups (*n* = 5 per group) and vaccinated intraperitoneally with a single dose of 1× 10^6^ CFU of A19 strain or A19mut2 according to the report [12,16,17]. An equal number of mice were given only PBS as unvaccinated controls. Protective efficacy induced by A19mut2 injection in BALB/c mice was measured and compared with that of A19 strain at 6 WPI. All groups of mice were challenged intraperitoneally at 6 WPI with 1 × 10^6^ CFU of S2308 Nal^R^ per mouse as reported [16]. Two weeks after the virulent challenge, the mice were sacrificed, and the spleens were collected, weighed, and homogenized in PBS. The numbers of CFU recovered from spleens were determined by plating TSA with 15 μg/mL nalidixic acid as described above. The levels of infection were expressed as the mean ± SD of log CFU/spleen of S2308 Nal^R^ after challenge. Mean log CFU reductions or log units of protection were obtained by subtracting the mean log CFU for the vaccinated group from the mean log CFU for the unvaccinated control group.

### 2.14. Statistical Analysis

Data were imported into GraphPad Prism 9.0 (Graph Pad Software, San Diego, CA, USA) for analysis. Statistical significance was determined using a Student’s *t*-test. For group analysis, one-way ANOVA or two-way ANOVA followed by Dunnett’s multiple comparisons test was used. *p* values less than 0.05 were considered statistically significant.

## 3. Results

### 3.1. A19-Derived Mutant A19mut2 Was Constructed Successfully with an Antigenically Modified O-PS

A19mut2 was derived from *B. abortus* vaccine strain A19, in which the *wbkC* gene was replaced by the *wbdR* gene from *E. coli* O157 using the method of gene homologous recombination (Figure 1A). Firstly, we verified the A19mut2 by PCR assay, showing that the fragment of the *wbdR* gene in A19mut2 was amplified using the cross primers ID-F1 and ID-R1, but not in A19 (Figure 1B). When the fragment of the *wbkC* gene was amplified by PCR using the cross primers ID-F1 and ID-R2, the target band was shown in the control strain A19, but not in the A19mut2 strain (Figure 1C). The result revealed that in the A19mut2 strain, the *wbkC* gene was successfully replaced by the *wbdR* gene. Secondly, the growth curve of A19mut2 was compared with that of A19, showing no significant difference (Appendix A). Then we analyzed the smooth phenotype of A19mut2 using acridine yellow agglutination and heat agglutination tests. The results showed that the A19mut2 did not agglutinate with 1% acridine orange or with heat incubation for 2 h (Figure 1D,E), which was similar to the behavior of the A19 strain. The rough-type *Brucella* control strain RB14 showed significant agglutination when treatedwith acridine yellow or heat (Figure 1D,E). The results revealed that A19mut2 is a smooth-type strain. Finally, to further confirm that A19mut2 expressed a modified O-PS that was distant from that of the original strain, A19, LPS silver staining and Western blotting were performed for verification. The LPS staining showed that both strains exhibited an integral LPS structure, but the migration profile of O-PS from A19mut2 was different from that of the A19 strain, displaying a shortened O-PS (Figure 1F). Additionally, Western blotting was carried out to evaluate the antigenicity of O-PS from A19mut2 using polyclonal antibodies against smooth *Brucella* and monoclonal antibodies specifically against O-PS, showing that the LPS from A19mut2 had no reaction with the polyclonal or monoclonal antibodies, but the LPS from the A19 strain had an obvious reaction with both antibodies (Figure 1F). These data suggested that the A19mut2 was successfully constructed with a smooth phenotype and showed an antigenically modified O-PS compared to that of the original strain, A19.

### 3.2. A19mut2 Reduced Its Ability to Resist Acidic pH, but Not Hydrogen Peroxide, Polymyxin B, and Sodium Nitroprusside (SNP)

To assess the ability of A19mut2 to resist bactericidal factors, hydrogen peroxide, SNP, polymyxin B, and an acidic buffer were used to evaluate its sensitivity compared to that of the A19 strain. As shown in Figure 2A,B, A19mut2 showed similar sensitivity to the hydrogen peroxide and polymyxin B. In the SNP test, A19mut2 also exhibited a similar sensitivity to SNP relative to A19 (Appendix A). These data suggested that the O-PS modification did not affect *Brucella*’s ability to resist killing by oxidative stress, cationic bactericidal peptides, and nitrosative stress. However, when incubated in a different acidic peptone buffer, A19mut2 had a significantly reduced survival rate at pHs of 6.5, 5.5, and 4.5 compared to that of the A19 strain (Figure 2C), suggesting that O-PS modification affects *Brucella*’s capacity to tolerate acid shock.

### 3.3. A19mut2 Showed Similar Residual Virulence Compared to the A19 Strain

To assess the residual virulence of A19mut2, BALB/c mice were inoculated by A19mut2 or A19 with a dose of 10^6^ CFU/mouse. The number of CFUs in the spleen was determined at 2, 4, 6, and 8 WPI. The results showed that the CFU of the A19mut2 recovered from the spleen was significantly reduced in comparison with that of the A19 strain at 2 WPI (Figure 3). However, although the bacterial loads recovered from the spleen inoculated by A19mut2 at 4 and 6 WPI were slightly higher than that of the A19 strain, no significant difference was found in both strains’ vaccinated groups. At 8 WPI, the A19mut2 showed a similar level of spleen colonization as the A19 strain (Figure 3).

To further evaluate the residual virulence of the A19mut2 strain, a histopathological examination via hematoxylin and eosin staining was performed in the spleens and livers inoculated by the A19mut2 and A19 strains. PBS inoculations were carried out as the negative control. The histopathology of the spleens vaccinated by A19mut2 showed the necrosis of white pulps and hyperplasia of connective tissues at 2 or 4 WPI, which were restored to the normal morphology at 6 and 8 WPI. The A19-inoculated spleen only showed obvious histopathological changes at 2 WPI, but not at 4, 6, or 8 WPI (Figure 4, left panel). The livers vaccinated by the A19mut2 and A19 strains showed visible granuloma at 2, 4, and 6 WPI and were restored to the normal morphology at 8 WPI with no obvious difference (Figure 4, right panel). Meanwhile, the PBS-inoculated spleens and livers showed no histopathological changes at all indicated times (Figure 4, right panel). The data suggest that the residual virulence of A19mut2 is basically equivalent to that of the A19 strain.

### 3.4. A19mut2 Induced Humoral- and Cellular-Mediated Immune Response

Sera were collected from the immunized mice at 2, 4, 6, and 8 WPI and assessed for the presence of *Brucella*-specific antibodies IgG, IgG1, and IgG2a based on the ELISA coated by the total *Brucella* S2308 lysates. The ELISA results showed that A19 induced significant levels of antibodies IgG, IgG1, and IgG2a in the mice compared to the PBS group (Figure 5). However, A19mut2 only induced significant levels of lgG and IgG2a in the mice at 6 and 8 WPI, but not at 2 and 4 WPI in comparison with the PBS group. Moreoever, the IgG1 level in the mice was not induced by A19mut2 in any of the phases of vaccination. Furthermore, when compared to the A19 vaccination, A19mut2 induced decreased levels of IgG, IgG1, and IgG2a in the mice (Figure 5). These data suggested that LPS is an important component in *Brucella* to induce antibody production, as previously reported in the literature [18]. Although the levels of antibodies were reduced in the A19mut2-vaccinated mice, the lgG2a levels were predominant in their immune responses, suggesting a bias towards to a Th1-type response.

To evaluate the cellular immune response, the supernatants were collected from the stimulated splenocytes of the mice at 6 WPI, and the Th1-type cytokine IFN-γ and the Th2-type cytokine IL-10 were assessed using an ELISA kit. As shown in Figure 6, when the splenocytes were stimulated by heat-killed S2308, the splenocytes from the A19- and A19mut2-immunized mice produced significant amounts of IFN-γ and IL-10 in comparison with that of the PBS-inoculated mice, and no significance was found between the A19 and A19mut2 groups in terms of cytokine production. The splenocytes in the RPMI1640 incubation produced low amounts of cytokines in all the tested groups. These data suggested that A19mut2 induced similar humoral and cellular immune responses with a predominately Th1-type response to the A19 strain.

### 3.5. Vaccination with A19mut2 Confers a Similar Protection to A19 Immunization in a Mice Model

To evaluate the protective efficacy of A19mut2, the mice were immunized with A19mut2 or A19, and one separate group was given PBS as a negative control. When the immunized mice were challenged by a wild-type S2308 Nal^R^ strain, both strains offered significant protection compared to that of the PBS-dosed mice and a significant decrease in the bacterial load in the spleens relative to that in the PBS-dosed mice, with a 1.19-log and 0.81-log reduction for A19 and A19mut2, respectively (Figure 7). Although the bacterial loads in the spleens of the A19mut2-immunized mice were a little higher than those of the A19 immunization, there were no significant differences in the bacterial loads between the A19- and A19mut2-immunized mice (Figure 7). These results indicated that the A19mut2 vaccination confers a similar protective efficacy in comparison with the A19 vaccination in mice.

### 3.6. A19mut2-Immunized Mice Can Be Discriminated from A19 Immunization via LPS-Coated ELISA

To evaluate if LPS could be used as a diagnostic antigen to distinguish A19mut2 and A19 immunization or wild-type *Brucella* infection, the sera were collected from the A19mut2- and A19-immunized mice at 2, 4, 6, and 8 WPI, and a wild-type LPS-coated ELISA was used to detect the antibody levels. The result showed that the antibody level of the A19mut2-immunized mice was significantly reduced relative to that of the A19-immunized mice based on the wild-type LPS-coated ELISA, which was similar to the antibody level of the PBS-dosed mice at all indicated times (Figure 8). These data suggested that the A19mut2-immunized mice could be successfully discriminated from the A19-immunized mice based on the wild-type LPS-coated ELISA.

## 4. Discussion

The O-PS of smooth-type *Brucella* LPS is an unbranched homopolymer of N-formyl-perosamine (4-formanido-4,6-dideoxy D-mannose) in various proportions of α-(1-2)- and α-(1-3)-linkages [19], which is exposed to the surface of *Brucella* to form the main epitopes in serological diagnoses. The genetic modification of *Brucella* O-PS could open the possibility of solving the diagnostic interference induced by current *Brucella* vaccines. In previous literature, Estrella M. et al. introduced an O-PS acetyltransferase WbdR from *E. coli* O157:H7 into *Brucella* to replace its O-PS formyltransferase WbkC, generating a *Brucella* mutant with an O-PS modification carrying N-acetyl-perosamine [20]. This ingenious strategy offers an effective solution to the current issue of serological differential diagnoses between vaccine immunization and wild-type *Brucella* infection. In this study, the *wbdR* gene was inserted into the genome of *B. abortus* vaccine strain A19 to replace the *wbkC* gene, producing a *B. abortus* vaccine mutant, A19mut2. As anticipated, A19mut2 exhibited a modified and shortened O-PS in comparison to that of the original O-PS, which was consistent with the previous *Brucella* construct, Rev 1::Tn7wbdRΔwbkC [6]. Moreover, in a previous study, *B. abortus wbdR* constructs (Ba::Tn7wbdR and Ba::Tn7wbdRΔwbkC) exhibited a more sensitivity to bovine serum, but not to bactericidal polycations, such as polymyxin B [20]. In this study, we showed that the sensitivity of the A19mut2 to polymyxin B remained unchanged compared to that of the A19 strain, which was in line with that of *B. abortus wbdR* constructs [20]. Interestingly, our study revealed that A19mut2 exhibited a greater sensitivity to low pHs compared to that of the A19 strain. It is well known that lysosomal acidification is required for the activation of the type IV secretion system and the subsequent intracellular survival of *Brucella* [21]. Although we did not find that A19mut2 reduced its intracellular survival within the RAW264 cells, different cells should be further evaluated, especially primary macrophages. Anyway, these data let us assume that the modified O-PS decreased the resistance of the *Brucella* to acidic environments or perhaps the resistance to natural serum, which may account for the attenuated virulence of A19mut2 in the early immune stage.

The protection by *Brucella* vaccines is mediated by the induction of humoral- and cellular-mediated immunity. In this study, the humoral and cellular immune responses induced by the A19mut2 strain were characterized by analyzing the antibody response and the cytokines released in the splenocytes induced by the immunogen, respectively. In the humoral immune response, although A19mut2 can induce higher levels of *Brucella*-specific IgG antibodies than in unvaccinated mice, the levels were significantly lower than those of the A19 induction. This reduction is most likely due to the O-PS modification; A19mut2 lost the ability to induce antibodies against the original LPS. Moreoever, the levels of specific IgG1 and IgG2a antibodies in serum suggested that the A19mut2 induced a Th1- and Th2-type mixed immunity and found that IgG2a predominated in the A19mut2-immunized mice, suggesting that their immunity was biased towards a Th1-type response. This result was similar to that previously reported for *B. abortus* BA15Δ*cydC*Δ*cydD* and BA15Δ*cydC*Δ*purD* mutants [14]. In terms of cellular immunity, the levels of cytokine secretion in the splenocytes induced by the A19mut2 strain were similar to those of the A19 strain. Meanwhile, it was shown that the Th1-type cytokine IFN-γ was more prominently secreted than the Th1-type cytokine IL-10. All these data suggested that vaccination with A19mut2 favors a Th1-biased humoral- and cellular-mediated immune response. A similar result was also found in other vaccine strains, such as *B. abortus* IVK15*ΔcydD* and IVK15Δ*cydC* mutants [22], and the *B. abortus* 2308Δ*nodV*Δ*nodW* double mutant [23]. Th1-biased immune response is deemed to play a major role in the establishment of a protective response against *Brucella* [24].

The protection rate of *Brucella* live vaccines is usually expressed by the reduction in the bacterial loads in the spleens of immunized mice via the wild-type *Brucella* challenges. Although the protection of A19mut2 was similar to that of the A19 strain with no significant difference, the bacterial loads in the mice’s spleens were higher in the A19mut2-vaccinated mice after the virulence challenge than those in the mice with the A19 vaccination. Interestingly, this phenomenon was also found in Rev1 wbdR tag vaccine Rev1::Tn7wbdRΔwbkC immunization against *B. ovis* [6]. When a higher dose is used in the inoculation, the vaccine can confer a similar protection to the Rev1 at the standard dose. It is worth noting that the property of autoagglutination suggests surface similarities between the Rev1::Tn7wbdRΔwbkC construct and rough-type *Brucella* [6]. However, we found that A19mut2 did not show an autoagglutination property like that of the rough-type *Brucella,* as shown above (Figure 1), which had properties biased towards smooth strains. Moreover, Rev1::Tn7wbdRΔwbkC displayed a reduced virulence compared to that of the Rev1 strain, but A19mut2 did not show a significant reduction in virulence relative to that of the A19 strain, especially in the late stage (Figure 3). It can be concluded that the modification of O-PS in different *Brucella* spp. may present different biological characteristics. Moreover, the antibodies anti-O-PS produced in the immunized mice may also confer protection against smooth-type *Brucella*, which can be concluded from the mice that were partially protected by inoculation with monoclonal antibodies against LPS [25,26]. The N-acetyl-perosamine O-PS in A19mut2 cannot induce mice to produce the antibodies against the original O-PS (N-formyl-perosamine), which may explain the slight reduction in protection relative to that of the A19 strain.

The great advantage of N-acetyl-perosamine replacing N-formyl-perosamine in O-PS is the possibility of differential diagnoses. This replacement of O-PS in *B. abortus* and *B. melitensis* was easy to distinguish based on ELISAs and Western blotting [20]. In this study, we evaluated the differential diagnoses between A19 and A19mut2 vaccinations in mice based on ELISAs coated by wild-type LPS. The result was unexpectedly attractive; the mice vaccinated with A19mut2 were significantly distinguished from the ones vaccinated with the A19 strain. Applications of this in clinical samples and groups need to be further evaluated.

## 5. Conclusions

In conclusion, a *B. abortus* A19-deprived mutant strain, A19mut2, was successfully constructed with the modification of the O-PS, which can induce significant humoral and cellular immunity in immunized mice with a bias towards to a Th1-type immune response, as the A19 vaccination showed. The protection efficacy of the A19mut2 vaccination in mice was similar to that of the A19 strain. Wild-type LPS-coated ELISAs distinguished the mice antibodies induced by the A19mut2 and A19 vaccinations. The results indicated that A19mut2 could be used as a DIVA vaccine candidate in the future.

## Figures and Tables

**Figure 1 vaccines-11-01273-f001:**
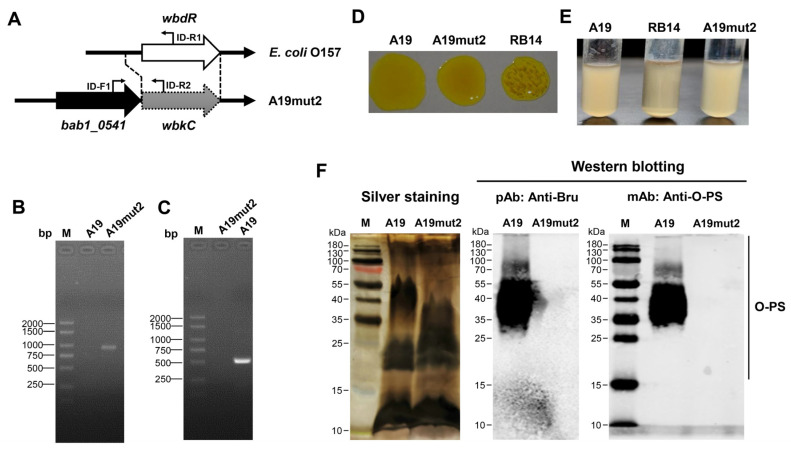
The construction strategy of A19mut2 and the modification of O-polysaccharide in A19mut2. (**A**) Schematic of the construction strategy of A19mut2; the *wbkC* gene was replaced by the *wbdR* gene from *E. coli* O157. (**B**,**C**) PCR identification of the A19mut2 strain using primer pairs ID-F1/ID-R1 (**B**) or ID-F1/ID-R2 (**C**). (**D**) Acridine yellow agglutination test of A19mut2, A19, and rough-type *Brucella* RB14. (**E**) Heat agglutination test of A19mut2, A19, and RB14. (**F**) Silver staining and Western blotting of the A19mut2 and A19 lipopolysaccharide. Rabbit anti-*Brucella* polyclonal antibodies (pAb) or mouse anti-O-PS antibodies (mAb, clone 51C) were used as primary antibodies.

**Figure 2 vaccines-11-01273-f002:**
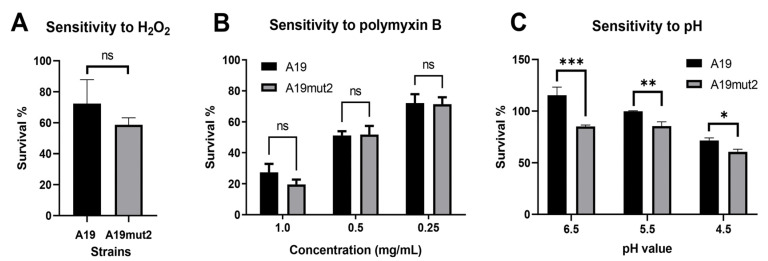
The sensitivity of A19mut2 and A19 strains’ bactericidal factors. (**A**) Sensitivity to hydrogen peroxide. (**B**) Sensitivity to polymyxin B. (**C**) Sensitivity to acidic environment. ns, no significance; *, *p* < 0.05; **, *p* < 0.01; ***, *p* < 0.001.

**Figure 3 vaccines-11-01273-f003:**
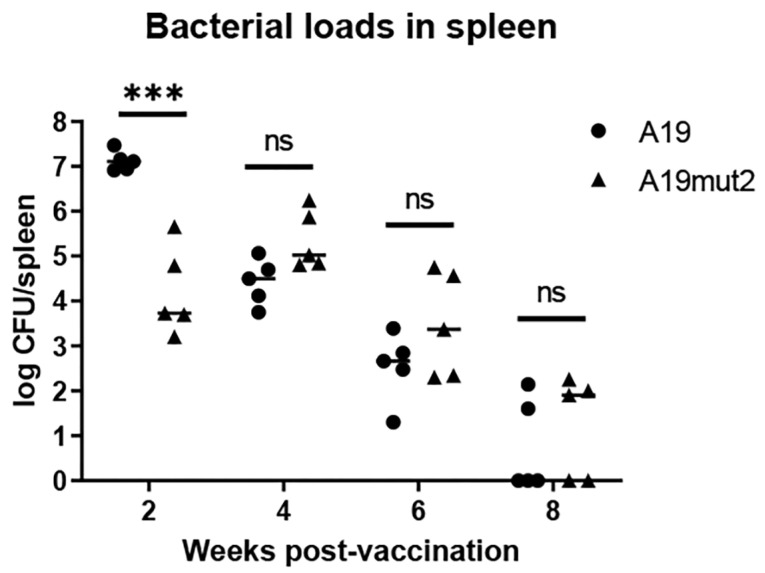
The residual virulence of A19mut2 was evaluated and compared with A19 in a mice model. Mice were intraperitoneally inoculated with 1 × 10^6^ CFU of A19mut2 or A19 strain, and the bacterial loads in spleens were enumerated at 2-, 4-, 6-, and 8-weeks post inoculation. ns, no significance; ***, *p* < 0.001.

**Figure 4 vaccines-11-01273-f004:**
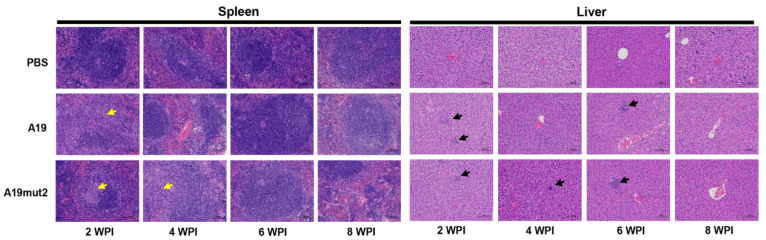
Histopathological changes by hematoxylin and eosin staining. Changes were observed in spleens and livers of mice inoculated with A19mut2, A19, and PBS at 2-, 4-, 6-, and 8-weeks post inoculation. The yellow arrows indicate white pulp necrosis and connective tissue proliferation. The black arrows indicate granulomas. Images were magnified 200×.

**Figure 5 vaccines-11-01273-f005:**
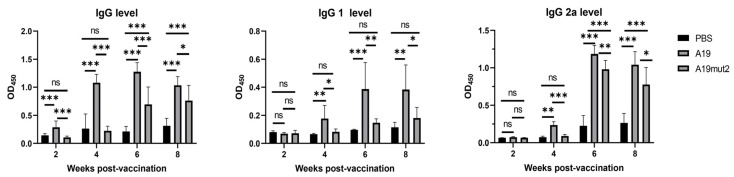
The humoral immune response was evaluated in immunized mice. The antibody levels of IgG and its subclass in sera of immunized mice were assessed using enzyme-linked immunosorbent assay coated by smooth-type *Brucella* lysates. ns, no significance; *, *p* < 0.05; **, *p* < 0.01; ***, *p* < 0.001.

**Figure 6 vaccines-11-01273-f006:**
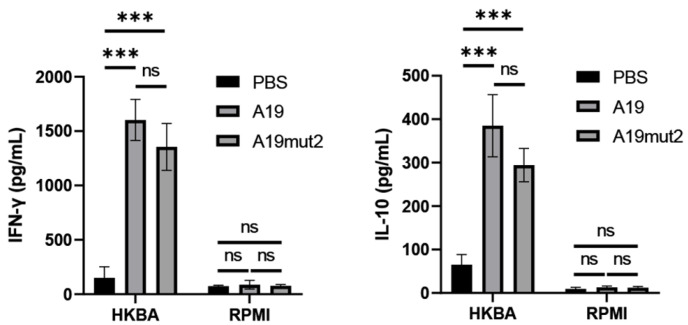
The cellular immune response was evaluated in immunized mice. Splenocytes were isolated from the mice at 6 weeks after vaccination by A19mut2, A19, or inoculation by PBS. The cytokines IFN-γ (A) and IL-10 (B) were detected in splenocytes stimulated by the heat-killed S2308 strain or incubated in RPMI1640. ns, no significance; ***, *p* < 0.001.

**Figure 7 vaccines-11-01273-f007:**
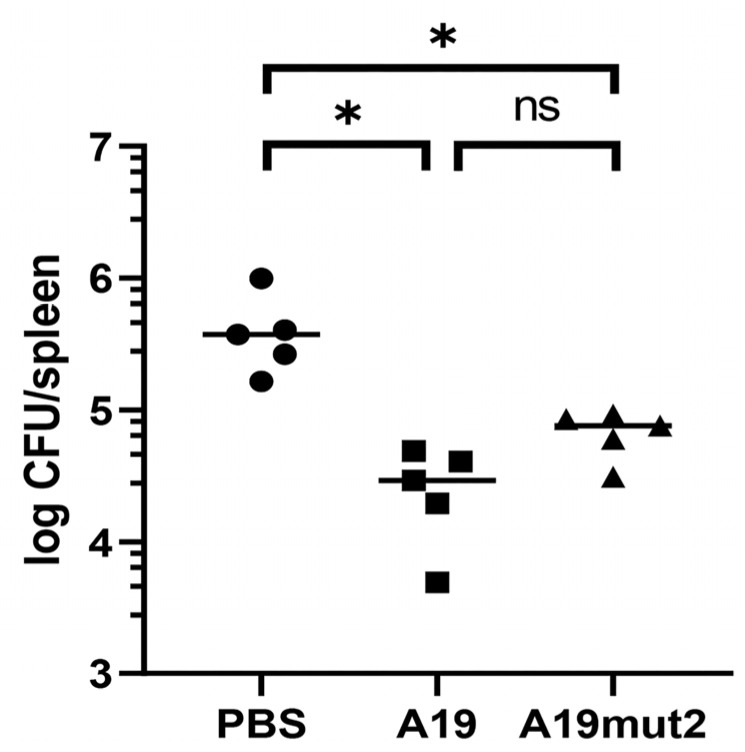
The protection of A19mut2 or A19 immunization against S2308 Nal^R^ challenge in a mice model. Bacterial loads in spleen of mice were calculated 2 weeks post challenge. ns, no significant; *, *p* < 0.05.

**Figure 8 vaccines-11-01273-f008:**
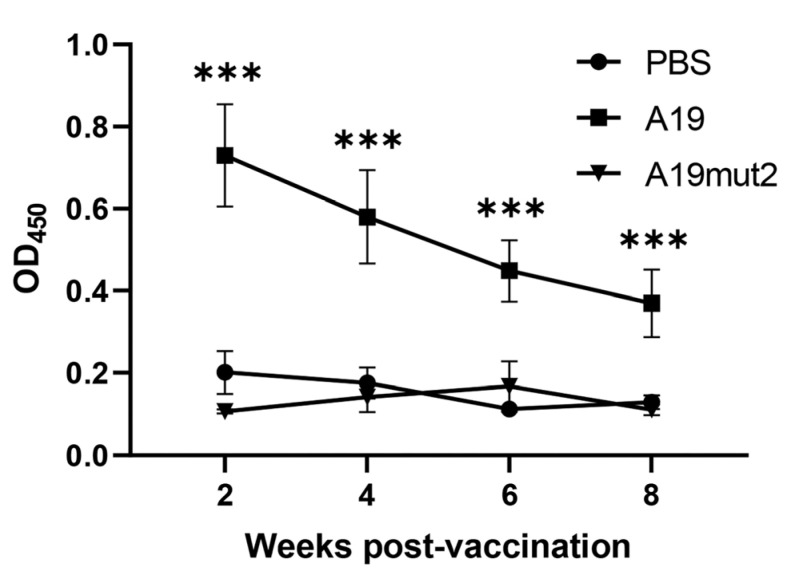
Discrimination of A19mut2 vaccinated mice sera from A19 via S2308-LPS-coated ELISA. The total IgG antibody levels were detected in mice sera at 2, 4, 6, and 8 weeks post immunization. ***, *p* < 0.001.

**Table 1 vaccines-11-01273-t001:** Bacterial strains, plasmids, and primers used in this study.

Names	Description	Source
Bacterial strains		
S2308	*B. abortus* wild-type strain 2308; smooth phenotype	CVCC ^a^
S2308 Nal^R^	*B. abortus* wild-type strain 2308 with nalidixic acid resistance; smooth phenotype	[7]
RB14	S2308 derived rough-type mutant with *rfbE* deletion; rough phenotype	[8]
A19	*B. abortus* vaccine strain A19; smooth phenotype	CVCC
A19mut2	A19 derived mutant with *wbkC* replaced by *wbdR*; smooth phenotype	This study ^b^
*E. coli* DH5α	F−φ80lacZ∆M15∆(lacZYA-argF)U169 recA1 endA1 hsdR17(rk−,mk+) phoA supE44 thi-1 gyrA96 relA1 λ−	TIANGEN ^c^
Plasmids		
pKB	pUC19-derived suicide plasmid containing *sacB* gene; KanR;	[9]
pKB-ΔwbkC::wbdR	pKB containing the upstream and downstream fragments of the *wbkC* gene; the *wbkC* gene was replaced by the *wbdR* gene	This study ^b^
Primers		
WbkC-F	GGTACCCGGGGATCCTGATGGCAGGGTAGAAGACG	This study ^d^
WbkC-R	TGCCTGCAGGTCGACCGTCTTCAAGTGCTGCTCAG	This study ^d^
rWbkC-R	TTAAAATGCCTCTTTTTCGTCA	This study ^d^
rWbkC-F	AATGGCTTGTTGCTTGTTTAGG	This study ^d^
WbdR-F	AAAGAGGCATTTTAAAGAAGTTCGCCACAGTAAATCGAA	This study ^e^
WbdR-R	AAGCAACAAGCCATTTTAAATAGATGTTGGCGATCTT	This study ^e^
ID-F1	GATCCCGGTTGTTGATGACG	This study ^d^
ID-R1	GCCCCAGGAGCAAATGTAAC	This study ^e^
ID-R2	GGAAGAAGCGACGGATGAAG	This study ^d^

^a^ The Chinese Veterinary Culture Collection Center, Beijing, China; ^b^ Bacterial strain and plasmid were constructed in this study; ^c^ TIANGEN Biotech Co., Ltd., Beijing, China; ^d,e^ The primers were designed based on the genome of *B. abortus* A19 strain (GenBank reference No. NZ_CP030751.1) ^d^ and on the genome of *E. coli* O157:H7 strain (GenBank reference No. NC_013008.1) ^e^ using the software Primer Premier 5.0 (PREMIER Biosoft, San Francisco, CA, USA).

## Data Availability

The datasets generated and analyzed during the current study are available from the corresponding author on reasonable request.

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
