# Peer review of "Characterization of Brucella abortus Mutant A19mut2, a Potential DIVA Vaccine Candidate with a Modification on Lipopolysaccharide"

_vaccines, 2023, doi:10.3390/vaccines11071273_

Round 1
Reviewer 1 Report
materials and methods are well detailed.their interpretation of the results is clear and convincing, with no comments that do not follow from their results. Este es un ejemplo Here is an example:
These data suggest that the A19mut2 strain was successfully constructed with a smooth phenotype and showed an antigenically modified O-278 PS compared to the original A19mut2 strain.
I agree with the first part of the discussion:Genetic modification of Brucella O-PS could open the possibility 410 of solving the diagnostic interference induced by current brucellosis vaccines.
In this part of the discussion I completely agree. It should be tested in bovines. It seems to me that it is ready for testing:
Anyway, the protective efficacy of the A19mut2 should be further investigated in natural host, like cows.
I consider this to be a repetition and should not be part of the discussion. It was already said in the introduction.
Diagnostic interference in Brucella vaccination is a major obstacle to the elimination 473 of brucellosis [31].

The English of the article is excellent, it seems to be written or corrected by someone who speaks and writes English like a native speaker.
Author Response
Anyway, the protective efficacy of the A19mut2 should be further investigated in natural host, like cows.
Reply: Thanks for your comment. We agree with your suggestion and will assess this DIVA vaccine in natural host in further study.
I consider this to be a repetition and should not be part of the discussion. It was already said in the introduction.
Diagnostic interference in Brucella vaccination is a major obstacle to the elimination of brucellosis [31].
Reply: Thanks for your comment. We have removed this sentence from the part of discussion in the revised manuscript.
Reviewer 2 Report
Dear Editor and authors
Authors in the current manuscript aimed to develop a Brucella vaccine candidate with further attenuated virulence. They constructed a B. abortus A19 derived mutant (A19mut2) by replacing the formyltransferase gene wbkC-20 associated with LPS synthesis by the acetyltransferase gene wbdR from Escherichia coli 21 O157, generating a modified O-polysaccharide. Then they performed characterization by PCR, tested its efficacy by comparing it with positive and negative controls. The work is very good and interesting. However, I have many comments (some are listed here and others in the manuscript):
Title:
- How it could be a differential diagnostic vaccine? this needs rephrasing.
Abstract:
- Subtitles (background, aim, results, and conclusion) should be added to be clear.
- Study design and method are missed.
- In line 16, however, both sentences have similar meanings. better to delete this word.
- Lines 22 to 24, the author mentioned " In further study, we found that the A19mut2 22 showed a reduced resistance to acidic pH and decreased its residual virulence for mice in early 23 inoculation stage in comparison with the A19 strain. " if this in another study, why it is mentioned in the abstract. better to delete this sentence.
- In line 30, is this the conclusion?? If so it is very short and differential diagnosis is thing and vaccination is another totally different thing, how can you bring them together in the same conclusion?
Introduction:
- It is very long, depends mainly on serodiagnosis and the hypothesis is not clear.
- A lot of studies were mentioned and these are better to be in the discussion part.
- In line 44, where is the reference for this, and this needs clarification as serodiagnosis depends mainly on Abs how their presence interferes with serodiagnosis.
- Line 48, the reference used is too old (2010), the levels of antibodies differ between the infected and vaccinated ones. The presence of Abs is the main point to test the efficacy of any vaccine.
Methods:
- References of the procedures are lacking.
- In table 1, the source for the primers used is the current study, how could your source is the current study, ref. should be used.
- Where is the reference for the infectious dose??
- Why you did not test for IgM, this indicates early infection or primary immune response not IgG?
- How did you decide on this dose?
Results:
- Mice immunized with A19 and A19mut2 showed similar results, however, in most results as cytokines, IgG A19 results were better.
-
Discussion:
- Other undated studies are needed to compare the results.
- More clarification of your results is needed.
Conclusion:
- the diagnostic role and vaccine role should be separated not mentioned together to clarify your idea. I think its role as a vaccine is minimal compared to the used one, particularly, a non-significant difference is noted.
References:
- References need to be updated, some references from 1986, 1989, 1990, 1992, 1993, 2006, 2009, …etc
- References 3 and 26 are the same.
- All references need to be checked for repetition.
- Self-citation is noted in ref. 11, 13, 14, 15, 17, 19.
Best regards,

The manuscript needs minor revision of the language.
Author Response
- Title:
How it could be a differential diagnostic vaccine? this needs rephrasing.
Reply: Thanks for your comments. The A19mut2 vaccine has a modified OPS that is different from the wild-type OPS of A19. ELISA coated by purified wild-type LPS can be used as a tool to distinguish A19mut2 immunized animals from A19 immunized ones. Following your valuable suggestion, we have modified the title in the revised manuscript to make it more clear.
- Abstract:
- Subtitles (background, aim, results, and conclusion) should be added to be clear.
Reply: Thanks for your comment. We have added subtitles in abstract section in the revised manuscript.
- Study design and method are missed.
Reply: Thanks for your comment. We have added study design and method in the revised abstract.
-In line 16, however, both sentences have similar meanings. better to delete this word.
Reply: Thanks for your suggestion. We have deleted the word in the revised manuscript.
-Lines 22 to 24, the author mentioned " In further study, we found that the A19mut2 showed a reduced resistance to acidic pH and decreased its residual virulence for mice in early inoculation stage in comparison with the A19 strain. " if this in another study, why it is mentioned in the abstract. better to delete this sentence.
Reply: Thanks for your comments. These experiments were performed in this study, not in further study. We have modified the description in the revised manuscript.
-In line 30, is this the conclusion?? If so it is very short and differential diagnosis is thing and vaccination is another totally different thing, how can you bring them together in the same conclusion?...”
Reply: Thank you for your insightful comments. Actually, in this study we developed a Brucella DIVA vaccine candidate. We are sorry for the ambiguity caused by this sentence, and modified it in the revised manuscript.
Introduction:
-It is very long, depends mainly on serodiagnosis and the hypothesis is not clear.
Reply: Thanks for your comment. We have shortened the introduction section to make it more concise.
-A lot of studies were mentioned, and these are better to be in the discussion part.
Reply: Thanks for your suggestion. We have revised it accordingly.
-In line 44, where is the reference for this, and this needs clarification as serodiagnosis depends mainly on Abs how their presence interferes with serodiagnosis.
Reply: Thanks for your comments. We have added the reference 3 in the revised manuscript. Interfere with serodiagnosis means that it is difficult to differentiate infected animal from vaccinated one using A19 vaccine by serological test. We have rephrased this sentence to make it more clear. The cited reference is: Yang, J.; et al. Evaluation and differential diagnosis of a genetic marked Brucella vaccine A19ΔvirB12 for cattle. Front Immunol. 2021, 12: 679560, 2021.
-Line 48, the reference used is too old (2010), the levels of antibodies differ between the infected and vaccinated ones. The presence of Abs is the main point to test the efficacy of any vaccine.
Reply: Thanks for your comment. The reference was updated in the revised manuscript. The A19 vaccine induces different levels of antibodies from the infected ones, thus based on the detection of antibodies against O-PS contributes to the current serodiagnosis of Brucella to some extent. However, A19 induced antibody do interfere serological diagnosis of Brucella clinically, which needs to be solved. In this study, we tried to modify OPS to reduce the interference of serodiagnosis, and it worked well. We agree that ”The presence of Abs is the main point to test the efficacy of any vaccine”, however, the protection efficacy of the Brucella A19mut2 is similar to that of A19 according to our results, suggesting the modification of OPS did not alter the protection efficacy of A19, thus A19mut2 could be used as a potential DIVA vaccine candidate.
Methods:
-References of the procedures are lacking.
Reply: Thanks for your comment. We have provided references in the revised manuscript.
-In table 1, the source for the primers used is the current study, how could your source is the current study, ref. should be used.
Reply: Thanks for the comment. These primers were designed by ourself for the current study, that’s why we mentioned “this study” in the table. We have added the notes in Table 1. The primers designed for construction of A19mut2 were mentioned in method section.
-Where is the reference for the infectious dose??
Reply: Thanks for the comment. We have added the reference in the revised manuscript.
-Why you did not test for IgM, this indicates early infection or primary immune response not IgG?
Reply: Thanks for your comments. Protection by Brucella live vaccines has been shown to be mediated by the induction of humoral and cell-mediated immune responses. In particular, a Th1-based cellular immune response plays a major role in the establishment of a protective response against Brucella. The levels of Brucella-specific IgG antibodies, and the subtype of antibodies IgG1 and IgG2a could reflect whether the response is a biased towards a Th1-type or Th2-type response. IgM also was induced in Brucella infection, which plays an important role in serological diagnosis of Brucellosis. In evaluation of Brucella vaccines, the determination of IgM titers was not commonly used as previous reports [14, 15, 16, 17, 22, 23]. So, in this study, we didn’t detect the IgM titers.
-How did you decide on this dose?
Reply: Thanks for your comment. The dose we used was according to the previous report, we cited the reference in the revised manuscript.
Results:
-Mice immunized with A19 and A19mut2 showed similar results, however, in most results as cytokines, IgG A19 results were better.
Reply: Thanks for your comment. In this study, we found that the A19mut2 can induce high levels of Brucella-specific IgG antibodies, but significantly lower levels than that of the A19. This is most likely due to the O-PS modification, the A19mut2 lost the ability to induce antibodies against the wild-type LPS. A19mut2 showed less cytokine production than A19, which may caused by its slight reduced virulence due to modified LPS in A19mut2. In cellular immunity, the levels of cytokine secretion in splenocytes induced by the A19mut2 strain were similar to that of the A19 strain. Meanwhile, it was shown that the Th1-type cytokine IFN-γ was more prominently secreted than the Th1-type cytokine IL-10. All these data suggested that the vaccination with the A19mut2 favors a Th1-biased humoral and cellular mediated immune response.
Discussion:
-Other undated studies are needed to compare the results.
Reply: Thanks for your comment. We have cited several recent studies and compared our findings with them in the revised manuscript.
-More clarification of your results is needed
Reply: Thanks for your comment. We have clarified all the results in our revised manuscript.
Conclusion:
-the diagnostic role and vaccine role should be separated not mentioned together to clarify your idea. I think its role as a vaccine is minimal compared to the used one, particularly, a non-significant difference is noted.
Reply: Thanks for your comments. As we expected, the protection efficiency of the A19mut2 was similar to that of the A19 vaccine. The main purpose of this study is to develop a potential DIVA vaccine candidate that can distinguish vaccinated animals from field virulent strain infected ones. The A19mut2 induced antibody in mice could be differentiated from A19 induced by a wild-type LPS-coated ELISA, suggesting it has great potential as a DIVA vaccine candidate in the future. Besides the differential diagnostic role of A19mut2, the vaccine role of protection is similar to that of A19 vaccine.
References:
-References need to be updated, some references from 1986, 1989, 1990, 1992, 1993, 2006, 2009, …etc
Reply: Thanks for your comment. we have updated these references in the revised manuscript.
-References 3 and 26 are the same.
Reply: Thanks for your comment. We have modified the references in the revised manuscript.
-All references need to be checked for repetition.
Reply: Thanks for your comment. We have checked all the references for repetition and made modifications in the revised manuscript.
-Self-citation is noted in ref. 11, 13, 14, 15, 17, 19.
Reply: Thanks for your comment. We have removed unnecessary self-cited references in the revised manuscript.
Reviewer 3 Report
Manuscript ID: Vaccines-2466966
Characterization of Brucella abortus mutant A19mut2, a potential differential diagnostic vaccine candidate with modification on Lipopolisaccharide.
In this work, a mutant designated A19mut2 was constructed and evaluated. It was demonstrated that the mutant may be a marker vaccine for brucellosis.
Merit:
The concept is good, and the methods employed are standard and straight forward. Development of a marker vaccine strain in infectious diseases research is a necessity.
Comments:
1. Introduction should be revised to make the manuscript attractive to read by the audience. Line 2 stated that “several trails have been performed”. I assume the authors meant “trials”. Please address. Lines 4-5 stated that …host animals, which interferes serodiagnosis” does not make sense. Please address. The last sentence needs revision.
2. Materials and methods: For bacterial growth determination (section 2.4) please indicate how many colony forming units (CFU) are in the OD600 of 0.1 and the equipment used to measure the optical density reading. This will confirm that an approximate number of A19 and A19mut2 strains in the starting inoculum was used. In section 2.9, please state how the 1 X 106 CFU/ml was confirmed.
3. The last sentence in the conclusion section should be re-written.
4. Aspects of the materials and methods section need improvement. Some areas lack specificity and clarity which will make reproducing the work by others challenging. As an example, comparative analysis of growth curve of two strains requires determination of actual CFU of both strains in the starting inoculum and standardization to remove bias. The authors also did not indicate the instrument/manufacturer used for growth curve analysis.
5. The manuscript is too long. Some aspects of the materials and methods should be eliminated and supporting articles referenced if published.
6. A problem that is consistent in the manuscript is proper construction of sentence making the manuscript not easy to read. This issue must be addressed.
7. Overall, the work is good and publishable if the concerns raised are properly addressed.
See comments
Author Response
- Introduction should be revised to make the manuscript attractive to read by the audience. Line 2 stated that “several trails have been performed”. I assume the authors meant “trials”. Please address. Lines 4-5 stated that …host animals, which interferes serodiagnosis” does not make sense. Please address. The last sentence needs revision.
Response: Thanks for your comment. We have revised the introduction to make the manuscript attractive to read by the audience. The typing error in line 2 was fixed as “trials”. We also revised the description of lines 4-5 and the last sentence to make these sentences more clear.
- Materials and methods: For bacterial growth determination (section 2.4) please indicate how many colony forming units (CFU) are in the OD600of 0.1 and the equipment used to measure the optical density reading. This will confirm that an approximate number of A19 and A19mut2 strains in the starting inoculum was used. In section 2.9, please state how the 1 X 106CFU/ml was confirmed.
Response: Thanks for your comment. We used an Eppendorf BioSpectrometer basic (Eppendorf, Hamburg, Germany) to measure OD600 value of the bacterial culture in TSB. The OD600 value of the starting inoculum was adjusted to 0.1 (about 5 × 108 CFU/mL). In section 2.9, the OD600 value of the bacterial culture was adjusted to 1 (about 5 × 109 CFU/mL), then 10-fold serially diluted to 5 × 106 CFU/mL (1,000 fold-dilution). Each mice were intraperitoneally inoluated with 0.2 mL (1 × 106 CFU). We have added the description in section 2.4 and 2.9 in the revised manuscript.
- The last sentence in the conclusion section should be re-written.
Reply: Thanks for your comment. The last sentence in the conclusion section has been revised in the revised manuscript.
- Aspects of the materials and methods section need improvement. Some areas lack specificity and clarity which will make reproducing the work by others challenging. As an example, comparative analysis of growth curve of two strains requires determination of actual CFU of both strains in the starting inoculum and standardization to remove bias. The authors also did not indicate the instrument/manufacturer used for growth curve analysis.
Reply: Thanks for your comment. We have revised materials and methods section, provided the information of actual CFU of both strains in the starting inoculum and SD value in the figure S1 in the revised manuscript. The information of the instrument/manufacturer used for growth curve analysis was also provided.
- The manuscript is too long. Some aspects of the materials and methods should be eliminated and supporting articles referenced if published.
Reply: Thanks for your comment. We have eliminated some aspects from the methods section, and more supporting articles were cited in the revised manuscript.
- A problem that is consistent in the manuscript is proper construction of sentence making the manuscript not easy to read. This issue must be addressed.
Reply: Thanks for your comment. We have carefully checked the manuscript for grammar and spelling errors, and thoroughly revised the whole manuscript with the help of our native English speaker colleague. We hope that the manuscript is now easy to understand.
- Overall, the work is good and publishable if the concerns raised are properly addressed.
Reply: Thanks for your comments. We have properly addressed all the comments raised by the reviewers.
Round 2
Reviewer 2 Report
Dear authors,
The corrections sound good, my only notice is in table 1, how is the source of primers designed is the current study?
Best regards,
Author Response
my only notice is in table 1, how is the source of primers designed is the current study?
Reply: Thanks for your comment. We have provided the sources of primers on the Table 1 footnote. In addition, we carefully checked the sequences of all the primers and revised several errors in the Table 1. The changes made in this version were highlighted in blue.